# The Impact of Foehn Wind on Mental Distress among Patients in a Swiss Psychiatric Hospital

**DOI:** 10.3390/ijerph191710831

**Published:** 2022-08-30

**Authors:** Christian A. Mikutta, Charlotte Pervilhac, Hansjörg Znoj, Andrea Federspiel, Thomas J. Müller

**Affiliations:** 1Private Clinic Meiringen, 3860 Meiringen, Switzerland; 2Translational Research Center, University Hospital of Psychiatry and Psychotherapy, University of Bern, 3000 Bern, Switzerland; 3Department of Physiology, Anatomy and Genetics, University of Oxford, Oxford OX1 3PT, UK; 4Institute of Psychology, Department of Health Psychology and Behavioural Medicine, University of Bern, 3012 Bern, Switzerland

**Keywords:** foehn wind, psychopathology, BSCL, mental health, weather, meteorological factors, climate change

## Abstract

Psychiatric patients are particularly vulnerable to strong weather stimuli, such as foehn, a hot wind that occurs in the alps. However, there is a dearth of research regarding its impact on mental health. This study investigated the impact of foehn wind among patients of a psychiatric hospital located in a foehn area in the Swiss Alps. Analysis was based on anonymized datasets obtained from routine records on admission and discharge, including the Brief Symptom Checklist (BSCL) questionnaire, as well as sociodemographic parameters (age, sex, and diagnosis). Between 2013 and 2020, a total of 10,456 admission days and 10,575 discharge days were recorded. All meteorological data were extracted from the database of the Federal Office of Meteorology and Climatology of Switzerland. We estimated the effect of foehn on the BSCL items using a distributed lag model. Significant differences were found between foehn and non-foehn admissions in obsession–compulsion, interpersonal sensitivity, depression, anxiety, phobic anxiety, paranoid ideation, and general severity index (GSI) (*p* < 0.05). Our findings suggest that foehn wind events may negatively affect specific mental health parameters in patients. More research is needed to fully understand the impact of foehn’s events on mental health.

## 1. Introduction

Global climate change has far-reaching implications, considering the effects that are already apparent and those that are to be expected in the future (i.e., IPCC [1]). In addition to the diverse ecological, political, socioeconomic, and demographic aspects associated with climate change, the individual effects of climate change on the population’s health must also be considered.

Public health research increasingly indicates the crucial and multifaceted impact of climate change on mental health and emotional well-being [2,3]. Recent literature reveals a correlation between experiencing climate change implications, such as extreme weather events or an increase in ambient temperature, and deterioration in mental health [3,4,5,6,7,8,9].

Data particularly show negative consequences related to ambient temperature and heat, with an increase in temperature being associated with an increase in mental distress, hospitalizations for mental disorders (including bipolar disorder, schizophrenia, and dementia) [8,10,11,12], and increased suicide rates [6,8,13,14,15,16].

In Switzerland, Bundo and colleagues [17] showed that the hospitalization risk for mental disorders increased linearly by 4.0% for every 10 °C increase in mean daily temperature in a 45-year study period. Higher risks have been identified in patients with developmental disorders and schizophrenia. Furthermore, heatwaves and extremely high temperatures have been associated with increased mortality in people suffering from mental illness [3,18,19,20,21,22]. Hence, research has shown that strong meteorological stimuli appear to exacerbate symptoms in people with mental disorder.

Despite the growing interest in the relationship between climate change events and mental well-being in public health research, little attention has been paid to weather conditions other than temperature—such as foehn [23]. Foehn wind is a well-known phenomenon in the Swiss Alpine region, generating a warm, dry, and often severe wind on the leeward downwind side of the mountain, descending from the crest into the valley, and reaching hurricane force wind speeds of 117 km/h (maximum exceeding 200 km/h) [24]. This meteorological phenomenon can be found all over the world in high mountain regions, such as ‘Halny’ in Poland or ‘Chinook’ in North America [25].

Although research remains scarce, foehn wind has been associated with a large variety of worsening health conditions, frequently labeled as ‘foehn illness’ [25,26]. These symptoms include migraine and headaches, joint pain, wound pain, sleep disorders, fatigue, concentration problems, anxiety, nausea, paresthesia, an increase in aggressiveness, agitation, depressive states, and suicide attempts [27,28,29,30,31,32,33,34]. Recently, Greve et al. [25] detected an increase in trauma-related hospital admissions on days with foehn. Some studies have found no effects on health conditions [29,35,36] or even relief of neuropathic pain during Chinook winds [37].

Bundo et al. [38] found that episodes of foehn were associated with a 5% increase in mental health hospitalizations over a 35-year study period in Switzerland. A recent study by Lickiewicz et al. [33] showed that lower barometric pressure and foehn wind increased aggressive behavior in psychiatric hospital patients in Poland. Despite these two studies, to the best of our knowledge, no other study has explored the impact of foehn on the mental distress of psychiatric inpatients—a particularly vulnerable group.

With mental illnesses affecting a billion people worldwide, causing 7% of the global burden of disease as measured in DALYs and 19% of all years lived with disability [39], mental disorders constitute a significant health and socioeconomic burden. The effects of climate change and people affected by mental illnesses are both projected to increase [3,40,41,42,43,44]. Therefore, a profound understanding of the potential risk factors related to mental disorders and the interplay between climate change and mental health must be strengthened to reduce this burden. Specifically, different psychopathological symptoms (and a combination thereof) seem to be appropriate targets for estimating the impact of foehn winds on mental health. However, to date, there are very limited data correlating specific psychopathology symptoms with foehn winds.

To relate self-reported clinically relevant psychological symptoms to the occurrence of foehn wind, we used the Brief Symptom Checklist (BSCL) in a sample of admissions and discharges from a psychiatric hospital located in a well-known foehn area in Switzerland. Specifically, based on the available literature, we expected foehn to increase ratings of aggressiveness and somatization.

## 2. Materials and Methods

The analysis was based on anonymized individual datasets obtained from routine ANQ records (Swiss National Association for Quality Development in Hospitals and Clinics) on the admission and discharge days of patients from the Private Clinic Meiringen, a psychiatric hospital located in a foehn area in the Swiss Alps. The collected data included the Brief Symptom Checklist (BSCL) questionnaire [45] and sociodemographic parameters (age, sex, and diagnosis). All admission and discharge data between 1st of January 2013 and 31st of December 2020 were included. During this interval a total of 10,456 admission and 10,575 discharge days were recorded. The difference in the number of admissions and discharges is the result of patients with pronounced psychopathology at the time-point of admission not being able to fill out the questionnaire. However, they were able to complete the questionnaire at the time-point of discharge. An exemption from the ethics committee of the University of Berne was obtained due to the anonymized data format.

### 2.1. Brief Symptom Checklist (BSCL)

The Brief Symptom Checklist (BSCL; known as the Brief Symptom Inventory (BSI) in English) is the short German version of the well-validated SCL-90 -R assessment designed by Derogatis [45,46,47]. The assessment is an important standardized multidimensional self-report scale designed to assess psychopathological symptoms and psychological distress experienced over the previous seven days. The BSCL is a 53-item instrument that measures nine dimensions: somatization (seven items), obsession–compulsion (six items), interpersonal sensitivity (four items), depression (six items), anxiety (six items), hostility (five items), phobic anxiety (five items), paranoid ideation (five items), and psychoticism (five items). These dimensions are assessed using three global indices: the General Severity Index (GSI), Positive Symptom Distress Index, and Positive Symptom Total. In our study, we used only the GSI as a global index. The original questionnaire reported an internal consistency coefficient range from 0.71 to 0.85 [48]. The test–retest reliability of the GSI after one week was shown to have a reliability of 0.87 [47]. Satisfactory convergent reliability has been reported for the depression subscale with Beck’s Depression Inventory [49], as well as a variety of other clinical questionnaires (r = 0.36, r = 0.83) [50].

### 2.2. Meteorological Data

All meteorological datasets were extracted from the database of the Federal Office of Meteorology and Climatology of Switzerland [51]. The location of the meteorological station is Meiringen (coordinates: 46°44′ N 8°10′ E at 588 m above mean sea level), in climate zone *Cfb* according to the Köppen and Geiger Classification (KGC) [52,53]. The observations from this station comprised the following variables: daily index of the presence of foehn, recorded every 10 min. The Foehn Index classifies the wind as follows: 0 (no foehn), 1 (moderate foehn), and 2 (strong foehn). Hence, the Foehn index was a categorical variable, with each day of the dataset represented by 144 Foehn index values.

To account for the fact that the Foehn index values have a different time scale to the BSCL data, we decided to co-register both datasets based on the BSCL data. Therefore, we assigned a single Foehn index value for one day. Specifically, although we had a total of 144 time points for the Foehn index, we compressed this information to one single Foehn index value for the whole day: 0 (no foehn), 1 (moderate foehn), and 2 (strong foehn). Therefore, we included 2922 days of meteorological data. Only a subsample of 259 days had a Foehn index value of 1 or 2, hereafter labeled as ‘Foehn events’.

### 2.3. Definition of the Foehn Group

The BSCL dataset included 10,456 admissions and 10,575 discharges. We identified all BSCL data of patients that fulfilled the following criteria: (1) present admission and discharge records have BSCL data, (2) admission and discharge BSCL records must be unique to one patient (i.e., no double or triple counting of the same patients), (3) Foehn index > 0 (i.e., foehn must be present). Using these criteria, 182 datasets were included in the foehn group.

### 2.4. Statistical Methods

The foehn group analysis was performed (a) comparing the foehn group to the admission group and (b) comparing the foehn group to the discharge group.To evaluate the potential short-term associations between Foehn index events and BSCL values, we first aggregated meteorological (Foehn index) data to BSCL data defined for all patients at admission and discharge. Therefore, the data retained the typical time-series structure. This model assumed stationarity in the association and aimed to control for the long-term effects of the Foehn index data on the BSCL data.We chose a distributed lag model (DLM) [54] to assess the potential impact of days characterized as ‘foehn’ on patients’ psychological distress symptoms, according to the BSCL, on hospitalization and discharge dates. The impact of this meteorological variable could be instantaneous and temporally lagged (e.g., Foehn index days over multiple previous days were used as predictors in a regression model for the specific outcome of interest).In the DLM, we included lags of the explanatory time-series as independent variables (i.e., Foehn index days). With this flexible framework, we aimed to compare different parametrizations of the exposure–response association and account for lagged dependencies. We used functions modeling the lag–response dimension with a maximum lag of seven days. Therefore, our model consisted of an unconstrained distributed lag linear model (i.e., assuming a linear association) with a maximum lag of seven days.

Moreover, we used generalized linear models (GLM) with conditional quasi-Poisson regression accounting for overdispersion, with the daily number of BSCL entities for each of that day’s patients as the dependent variable. Each day with events (i.e., Foehn index values > 0) was matched with the BSCL values for each patient. To evaluate the dependency of both variables, we used a ‘finite impulse response’ model within the GLM framework. For each Foehn index event, we constructed an event convolution matrix (ECM) that would account for potential lag dependencies. Therefore, this ECM consisted of a 7 × 7 matrix, with columns representing the time lag. For each row we placed a ‘1’ that was shifted for each column to the right (this is performed with the commands: eye (7) in MATLAB or diag(7) in R). This method does not make any assumptions regarding the exact shape of the response function.

Finally, we computed the paired Student’s *t*-tests against zero to identify significant time periods within each BSCL variable. *p*-values were corrected using false discovery rate correction (FDR). All statistical analyses were conducted in R (version 4.1.1 [55]) using the package ‘dLagM’ (of Haydar Demirhan) and using MATLAB ‘glmFit’ (MathWorks, Natick, MA, USA). It should be noted that we decided to display not only the potential lag dependencies after the events with Foehn index values of >0, but also to show the effect of this dependency for four days before this event in the respective figures.

## 3. Results

### 3.1. Description of the Foehn Group and Non-Foehn Group (Total Admission and Total Discharge Sample)

In the foehn group (n = 182), 47% of the patients were female and 52% were male. There was no significant difference between the sexes (*p* = 0.35). In the non-foehn admission group (n = 10,274), 52.1% of the patients were female, and 47.9% were male. In the non-foehn discharge group (n = 10,575) 52% of the patients were female and 48% were male. There was a significant difference in sex (*p* < 0.001) between the non-foehn’s admission and discharge groups. The Chi^2^ test revealed no significant sex difference between the foehn and non-foehn groups (Chi^2^ = 1.303, *p* = 0.520; Chi^2^ = 1.307, *p* = 0.521).

The mean age of the foehn group was 47.9 (SD: 13.2, range: 18–78), while the mean age of the non-foehn group was 47.86 (SD: 15.5, range 16–95) for the admission group and 47.80 (SD: 15.6, range 16–95) for the discharge group. No significant differences were found in the age of the groups (*p* > 0.05 for all).

Table 1, Table 2 and Table 3 depict the most common ICD-10 diagnosis codes found in the foehn group and the non-foehn admission and discharge groups, respectively. The Chi^2^ test did not reveal significant differences in the percentages of ICD-10 codes between the foehn group and the admission and discharge samples (Chi^2^ = 209.4, *p* = 0.582; Chi^2^ = 212.4, *p* = 0.574). Appendix A provides a comprehensive overview of all present ICD-10 diagnoses codes present in the sample.

### 3.2. Distributed Lag Model of foehn

First, we calculated the GLM with time lags to estimate the impact of foehn on the mental health variables of the BSCL. Figure 1 depicts the unconstrained distributed lag linear model (i.e., with a maximum lag of seven days).

### 3.3. T-Test of the BSCL Variables vs. Zero

Finally, we identified significant time lags for each BSCL variable. We found significant results on the following scales when comparing the foehn group with the admission sample: obsession–compulsion (t_max_ = 2.67, *p* = 0.010; time lag = day 2), interpersonal sensitivity (t_max_ = 2.59, *p* = 0.012, time lag = day 2), depression (t_max_ = 2.30, *p* = 0.027, time lag = day 3), anxiety (t_max_ = 2.32, *p* = 0.026, time lag = day 3), phobic anxiety (t_max_ = 3.00, *p* = 0.004, time lag = day 3), paranoid ideation (t_max_ = 2.20, *p* = 0.034, time lag = day 2), and GSI (t_max_ = 2.24, *p* = 0.031, time lag = day 3). Figure 2 depicts the development of the t-values (vs. zero) as a function of time lags for each BSCL variable with significant results. Appendix A shows the non-significant BSCL items of the admission group.

When comparing the foehn group with the discharge sample, we found significant results exclusively during negative time lags in obsession–compulsion (t_max_ = 2.03, *p* = 0.048; time lag = day-2), depression (t_max_ = 2.15 *p* = 0.037, time lag = day-2), and psychoticism (t_max_ = 2.29 *p* = 0.027, time lag = day-2) (see Appendix A).

## 4. Discussion

This study investigated the impact of foehn wind events on the mental distress of patients in a psychiatric hospital in Switzerland. Our results suggest that foehn winds might increase distress, negatively impacting the psychological state of patients with mental disorders before obtaining inpatient psychiatric treatment. These patients might be considered unstable, leading to hospitalization in psychiatric clinics. Thus, they appear particularly vulnerable to external stressors, such as meteorological factors, including foehn winds. Moreover, our findings indicate that at the time of admission, foehn episodes might have a negative influence on several specific psychopathological dimensions, namely, obsession–compulsion, interpersonal sensitivity, depression, anxiety, phobic anxiety, and paranoid ideation.

The negative health effects of foehn events at admission generally appeared to have a lag effect of two days, replicating similar lag effects of previous studies which examined the relationship between weather events and mental health outcomes [17].

Our results are generally in line with the–thus far, limited–literature, indicating that the foehn wind might represent a risk factor for particularly vulnerable groups. Bundo et al. [38] showed that foehn episodes led to a higher risk of mental health hospitalizations in Bern. Lickiewicz et al. [33] demonstrated an increase in aggressive behavior among psychiatric inpatients in Poland relating to foehn winds and barometric pressure changes.

Interestingly, no significant results after the foehn wind on mental distress outcomes were observed on the day of discharge. We theorize that patients might have acquired some stability in the course of their inpatient treatment and thus are less affected by external environmental stressors such as meteorological stressors. The shape of the potential dependency of all BSCL psychopathology variables and foehn events on the day of discharge suggests a stochastic process. In fact, it resembles a zig-zag curve with arbitrary significant negative time lags for obsession–compulsion, depression, and psychoticism. Therefore, the mental distress outcome at admission, measured by the BSCL variables (compared to discharge), may be regarded as a sensitive marker that links subjective symptoms to objective meteorological measures.

The biological mechanisms underlying the influence of meteorological factors on mental health remain largely unknown. Regarding heatwave-related exacerbations, several postulations have been put forth regarding the failure of thermoregulation, leading to an imbalance of serotonin levels, neuroinflammation in the brain, neurotoxicity, and alterations in metabolic functions [56,57,58]. Additionally, the dysfunctional effects of certain psychiatric medicines inhibiting adaptive thermoregulation of the body, a deficit of weather-dependent behavioral adaptations by people suffering from mental disorders, and disrupted sleep have been proposed [10,21,22,58,59]. However, to date, no specific pathophysiological hypothesis concerning foehn winds and mental health outcomes has been proposed.

A strength of this study is that, to the authors’ knowledge, no other study has explored the relationship between foehn episodes on specific psychopathological dimensions of people affected by mental disorders, thus, for the first time enabling insight into what particular dimensions of psychological distress (e.g., depression, anxiety, or somatization) might be negatively affected by foehn winds. Contrary to recent literature, our results suggest that somatization and aggressiveness are not negatively affected by foehn winds. However, in line with Bundo et al. [38] our data suggest that symptoms of depression and anxiety are negatively altered by foehn winds. This is of particular importance since anxiety/fear-related and depressive disorders are the most prevalent mental disorders worldwide [60,61,62].

Finally, with obsession–compulsion, interpersonal sensitivity, phobic anxiety, and paranoid ideation, further psychopathological entities were connected to foehn winds. Of note is paranoid ideation, since patients with psychotic symptoms appear to be particularly vulnerable to the influence of environmental factors [17], and so often need hospitalization in situations with symptom exacerbation.

The literature on climate change and mental health has been limited by research gaps and various limitations [3,23,63]. Therefore, the importance of studying climate-vulnerable groups is emphasized [3,23,63]. Another strength of the current study, therefore, lies in the investigation of a climate-vulnerable and marginalized group of psychiatric inpatients [23]. People suffering from mental disorders have been postulated to be particularly at risk for the effects of climate change on physical as well as mental health [3,18,64]. Extreme weather events elevate the risk of symptom exacerbation (cf. Lõhmus [58]), suicide [8,13,14,15], and a three-times-higher risk of mortality during heatwaves compared to people without mental disorders [18]. Other marginalized groups constitute women (especially pregnant or postnatal women), children, groups with less social support, insufficient medical care, people suffering from financial difficulties, and minoritized ethnic groups [3,23,65,66,67,68,69,70].

Regarding limitations, the study did not encompass confounding effects, for example, simultaneous exposure to intense foehn wind noise, with chronic noise having been identified as leading to negative mental health outcomes [71,72,73,74], and temporal noise exposure leading to feelings of annoyance [75]. Additionally, the sudden rise in ambient temperature is a phenomenon of foehn and was not considered in our study. However, it should be noted that the rise in mental health hospitalizations related to foehn winds over a 35-year study period has been shown to be independent of temperature [17].

As this study was conducted in a high-income country, looking at inpatient data, the interpretability and generalizability of our findings are somewhat limited [23].

Another limitation lies in the individual-level variability in foehn exposure, which confines the validity and precision of the exposure assessment of our population. Additionally, following a retrospective analysis, the study design did not allow for causal assumptions, such as pathophysiological mechanism between foehn and psychological distress. In the future prospective, personal exposure studies based on meteorological and geographical high-resolution data should be used in the risk assessment of weather stimuli.

As foehn winds are a frequent phenomenon of the Swiss Alps, affecting a large number of people, our results indicate the significance of future research examining the specifics of foehn winds. As such, it is critical to comprehend if (and why) certain foehn-affected communities and individuals demonstrate resilience in the light of foehn events.

Preliminary clinical prevention efforts and public health implications should consider building climate resilience by, among other things, increasing knowledge related to the climate change-related mental health risks for clinicians, individuals with mental health disorders, as well as their relatives or carers, using psychoeducational approaches to increase patients’ knowledge and insight of their mental disorder, including climate-related risk factors, constructing specific (behavioral) adaptation interventions and evaluating these empirically, and, importantly, identifying protective factors and coping mechanisms related to foehn winds, including learning from affected foehn communities [3,23,63].

## 5. Conclusions

Significant and positive associations were found between foehn wind events and specific psychological distress dimensions among patients at a Swiss psychiatric hospital at the time of admission. However, at the end of inpatient treatment, foehn wind events did not appear to represent a significant risk factor for patients. These results suggest that external meteorological stressors, such as foehn winds, could alter the mental status of the most vulnerable populations. If climate change and the number of climate-related stressors continue to increase, even more people with mental health risk factors might be affected by this development.

Thus, a profound understanding of the possible risk as well as protective factors related to mental health disorders and the interplay between climate change and mental health must be strengthened to reduce the burden of mental health disorders in the future.

## Figures and Tables

**Figure 1 ijerph-19-10831-f001:**
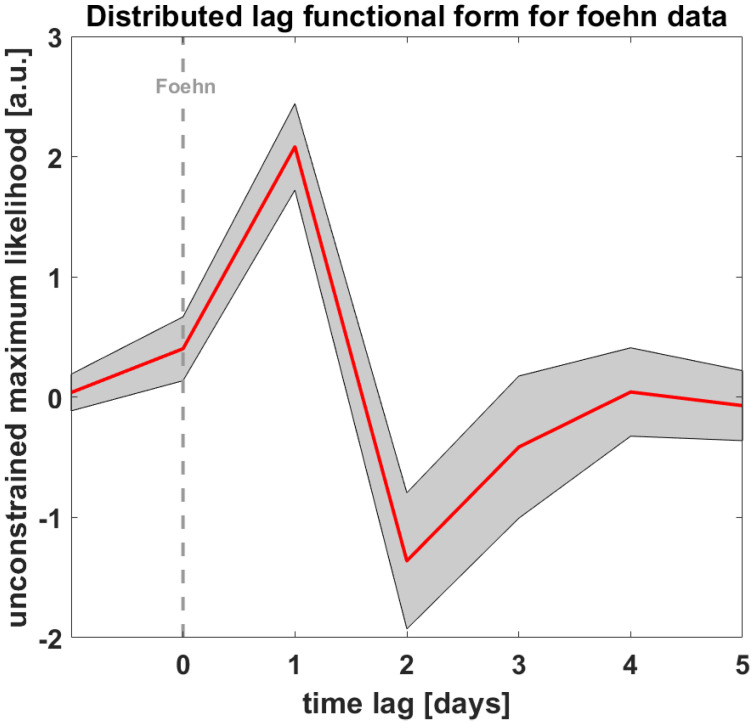
Depicts the distributed lag functional form for the foehn data. Zero equals the foehn day.

**Figure 2 ijerph-19-10831-f002:**
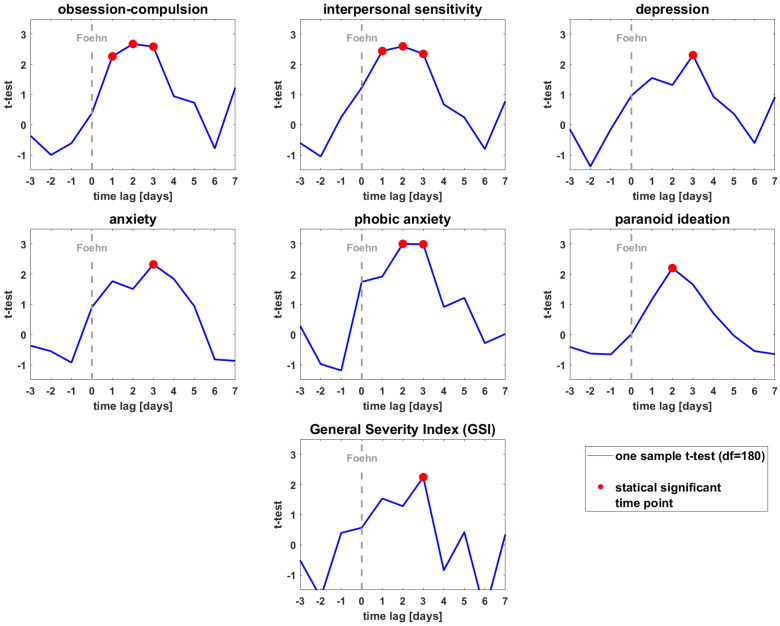
Depicts the t-values of the different significant BSCL variables (at admission) as a function of the time lags. Red dots indicate the significant time lags (*t*-test vs. zero *p* < 0.05).

**Table 1 ijerph-19-10831-t001:** Frequencies of ICD Codes in the foehn group.

ICD Code	Frequency	Percent
F32.1	56	30.8%
F33.1	40	22%
F10.2	19	10.4%
F33.2	13	7.1%

**Table 2 ijerph-19-10831-t002:** Frequencies of ICD Codes in the non-foehn admission group.

ICD Code	Frequency	Percent
F33.1	2001	19.1%
F32.1	1792	17.1%
F33.2	1210	11.6%
F10.2	1079	10.4%

**Table 3 ijerph-19-10831-t003:** Frequencies of ICD Codes in the non-foehn discharge group.

ICD Code	Frequency	Percent
F33.1	1996	19.2%
F32.1	1792	17.1%
F33.2	1210	11.6%
F10.2	1087	10.4%

## Data Availability

Data are available from the authors with reasonable request.

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
