# Peer review of "The Impact of Foehn Wind on Mental Distress among Patients in a Swiss Psychiatric Hospital"

_ijerph, 2022, doi:10.3390/ijerph191710831_

Round 1

Reviewer 1 Report

Thank you for the opportunity to review the article “The Impact of Foehn Wind on Mental Distress Among Patients in a Swiss Psychiatric Hospital.” The article is interesting and engaging, analyzing “the impact of foehn wind among patients of a psychiatric 14 hospital located in a foehn area in the Swiss Alps” (lines 14-15). This study fills a gap in the literature on the influence of weather phenomena on mental health, specifically on people suffering from certain mental disorders. In a context where climate phenomena have increased in recent years and experts warn that they will continue to do so, these studies are important to take into account and take measures to address their impact on vulnerable groups.

Overall, the subject is corrected attributed to the Mental Health section of the International Journal of Environmental Research and Public Health journal. The subject of the article is of interest to this field, and the perspective approached by the authors is innovative as they explore the associations between foehn wind events and specific psychological distress dimensions among patients at a Swiss psychiatric hospital at the time of admission. The authors used a dataset that included 10,456 admissions and 10,575 discharges; valuable entries to obtain relevant results.

Foremost, it is important to emphasize that the article is well-structured, the arguments are very clearly presented, and they are logically connected with the aim posed in the article's abstract. 

Moreover, the theoretical background and the literature review are appropriate for this subject and the presented arguments. The research design is consistent with the objective of the study and the data used.

I would also like to acknowledge and appreciate that the authors sought and received the exemption of the ethics committee of the University of Bern for their research.

The research results are clear and in line with the objective of the study. I also appreciate the perspectives outlined in the Discussion section and the fact that the authors have documented the limitations of their research very well.

I congratulate the authors and consider that the study in its present form can be published in the journal.

Author Response

Dear Reviewer, tank you very mach for providing this insightful and accurate review.Please find attached a letter with a point by point reply to your inquiries. 

Best regards

Christian Mikutta (for the authors)

Reviewer 2 Report

Line 85 - spacing between "land.Specifically"

Line 94 - please explain more discharges than admissions

Line 127 - was this over the duration of the study? Please clarify.

Line 137 - what is "John Group"?

Line 178 - spacing between 52 and % sign

Line 203 - center align figure-1 and caption

Line 215 - center align figure-2 and caption

Line 238 - awkward, rephrase "an increase in aggressive among"

Line 245 - incomplete "suggest a stochastic."

Line 296 - valid point - agreed

Line 298 - Casual assumptions such as? Please give example.

Line 320 - perhaps need to studied more to establish if "foehn winds, could negatively affect" 

Line 322 - "Climate-related stressors will increase in the future" any scientific meteorological prediction of this statement? Is it event or event' media exposure?

Author Response

(The authors gave the same response as above.)
